# Pregnancy and Lactation-Associated Osteoporosis Successfully Treated with Romosozumab: A Case Report

**DOI:** 10.3390/medicina59010019

**Published:** 2022-12-22

**Authors:** Yoichi Kaneuchi, Masumi Iwabuchi, Michiyuki Hakozaki, Hitoshi Yamada, Shin-ichi Konno

**Affiliations:** 1Department of Orthopaedic Surgery, School of Medicine, Fukushima Medical University, 1 Hikarigaoka, Fukushima City 960-1295, Fukushima, Japan; 2Department of Orthopaedic and Spinal Surgery, Aizu Medical Center, Fukushima Medical University, Aizuwakamatsu City 969-3492, Fukushima, Japan; 3Higashi-Shirakawa Orthopaedic Academy, School of Medicine, Fukushima Medical University, 1 Hikarigaoka, Fukushima City 960-1295, Fukushima, Japan; 4Department of Musculoskeletal Oncology and Metabolic Bone Disease Research, School of Medicine, Fukushima Medical University, 1 Hikarigaoka, Fukushima City 960-1295, Fukushima, Japan

**Keywords:** romosozumab, premenopausal osteoporosis, vertebral fracture, DXA, pregnancy and lactation, PLO

## Abstract

Pregnancy- and lactation-associated osteoporosis (PLO) is a rare type of premenopausal osteoporosis that occurs mainly in the third trimester or immediately after delivery; one of its most common symptoms is back pain caused by a vertebral fracture. The pathogenesis of PLO is unclear, and there is no accepted consensus regarding the treatment of PLO. Although treatments with drugs such as bisphosphonate, strontium ranelate, denosumab, and teriparatide were reported, there is no report of a patient with PLO treated with romosozumab. We present the first case of a patient with PLO treated with romosozumab following 4-month teriparatide treatment. A 34-year-old primiparous and breastfeeding Japanese woman experienced severe low back pain 1 month postdelivery. She was diagnosed with PLO on the basis of low bone marrow density (BMD) and multiple vertebral fractures with no identified cause of secondary osteoporosis. She was treated with teriparatide injection for 4 months, but the treatment was discontinued because of the patient feeling severe nausea after every teriparatide injection and the appearance of new vertebral fractures. Thereafter, we used romosozumab for 12 months. After the romosozumab treatment, her BMD was increased from the baseline by 23.6% at L1–L4, 6.2% at the femoral neck, and 11.2% at the total hip. Treating PLO with 12-month romosozumab after 4 months of teriparatide injection remarkably increased the BMD of the lumbar spine, femoral neck, and total hip without subsequent fracture. Romosozumab has potential as a therapeutic option to improve the BMD and reduce the subsequent fracture risk of patients with PLO.

## 1. Introduction

Pregnancy- and lactation-associated osteoporosis (PLO) is a rare type of premenopausal osteoporosis that occurs mainly in the third trimester or immediately after the delivery [1,2,3]. The pathogenesis of PLO remains unclear, and there is no accepted consensus regarding the treatment of PLO.

A common first step in the treatment of a patient with PLO is delactation with supplementation of calcium with or without vitamin D [1]. Various drugs were reported as treatment for PLO, including bisphosphonate [2,4,5], strontium ranelate (SrRan) [6,7], denosumab [8,9], and teriparatide [10,11,12]. Monoclonal antibody romosozumab, a sclerostin inhibitor, is one of the key drugs used to treat postmenopausal osteoporosis [13,14], but our search revealed no report of a patient with PLO treated with romosozumab. We present the first case of a patient with PLO treated with romosozumab.

## 2. Case Presentation

A 34-year-old primiparous and breastfeeding Japanese woman felt low back pain (LBP) 1 month after her infant’s delivery by caesarean section. The patient had no history of trauma. She was initially treated at a nearby clinic without any analgesic drug, but the LBP gradually worsened, and the patient presented to our hospital 3 months after the delivery. The severity of pain according to a numeric rating scale (NRS) was 9/10. The patient had undergone a laparoscopic partial oophorectomy for the treatment of endometriosis. She did not smoke and had no history of alcohol consumption, and there was no history of fractures. Her mother had osteoporosis without fracture. On clinical examination, the patient’s height was 160 cm, her weight was 50.2 kg, and her body mass index (BMI) was 19.6 kg/m^2^.

Plain radiographs of the lumbar spine showed a loss of height at the L1–L4 vertebrae (Figure 1A,B). Nonenhanced lumbar magnetic resonance imaging (MRI) revealed low intensity on T1-weighted imaging (T1-WI) and high intensity on short tau inversion recovery (STIR) imaging in the compressed vertebral body at L1–L4 (Figure 1C,D). The patient was diagnosed with vertebral fractures at L1–L4. Laboratory examination revealed alkaline phosphatase 109 IU/L (normal range, 38–119 IU/L), serum calcium 9.3 mg/dL (8.8–10.1 mg/dL), serum phosphorus 4.6 mg/dL (2.7–4.6 mg/dL), parathyroid hormone (PTH) 24 pg/mL (10–65), parathyroid hormone-related hormone (PTHrP) < 1.0 pmol/L (<1.1 pmol/L), thyroid-stimulating hormone (TSH) 0.27 μU/mL (0.61–4.23 μU/mL), prolactin 51.7 ng/mL (3.12–29.32 ng/mL), and 25 hydroxy vitamin D [25(OH)D] 13.6 ng/mL (>30 ng/mL). The bone turnover markers were also examined; procollagen type Ⅰ N-terminal propeptide (PⅠNP) 76.1 ng/mL (normal range 16.8–98.2 ng/mL) and tartrate-resistant acid phosphatase-5b (TRACP-5b) 476 mU/dL (120–420 mU/dL) (Table 1). The results of other serum biochemical examinations, including renal and liver function tests, were within the normal ranges.

The patient’s bone mineral density (BMD) was measured using dual-energy X-ray absorptiometry (DXA). As shown in Table 2, the respective BMD values and Z score of the lumbar spine (L1–L4) were 0.852 g/cm^2^ and −2.1; those of the right femoral neck were 0.710 g/m^2^ and −1.5; those of the total hip were 0.726 g/cm^2^ and −1.7. Because there was no cause of secondary osteoporosis, the patient was diagnosed with PLO. She stopped breastfeeding, and started to use a biweekly subcutaneous injection of teriparatide (56.4 μg/week) and oral supplementation of eldecalcitol (0.75 μg/day). One week after the initiation of treatment, hyperphosphatemia (6.2 mg/dL) occurred, and eldecalcitol was thus discontinued; the teriparatide injections continued.

To control her back pain, the patient started taking loxoprofen and used a thoracolumbar orthosis for 3 months, which gradually led to pain relief (NRS 2). At the 4th month of teriparatide treatment, the patient’s back pain suddenly worsened (NRS 4) when she was holding her baby, and additional radiological examinations were performed. Plain radiographs of the lumbar spine showed a slight loss of height at the Th11 vertebra (Figure 2A). MRI of the thoracolumbar spine detected low signal intensity on T1-WI and high signal intensity on STIR at both the Th11 and L5 vertebral bodies (Figure 2B). We diagnosed new vertebral fractures at Th11 and L5. Additional DXA measurements showed that the BMD was 0.842 g/m^2^ at the lumbar spine (L1–L4), 0.723 g/m^2^ at the femoral neck, and 0.721 g/m^2^ at the total hip.

To exclude Cushing’s syndrome, the patient’s serum cortisol and adrenocorticotropic hormone were measured, and both were within the normal range. Her PⅠNP level had increased to 113 ng/mL, and TRACP-5b had decreased to 348 mU/dL (Table 1). Because (i) new vertebral fractures had occurred, (ii) the BMD values had not improved, and (iii) the patient reported feeling severe nausea after every teriparatide injection, the teriparatide treatment was discontinued. After a thorough explanation and discussion, treatment with romosozumab along with calcium and eldecalcitol was started. Two months after the start of the romosozumab treatment, the patient’s LBP was remarkably improved (NRS 0), and increases in the BMD values at the lumbar spine, femoral neck, and total hip were observed (Table 2). Additional MRI described the improvement of signal change on Th11 and L5 vertebrae (Figure 2C,D). After 12 months of the romosozumab treatment, compared to the baseline values, the BMD values had increased by 23.6% at the lumbar spine (L1–L4), 6.2% at the femoral neck, and 11.2% at the total hip.

## 3. Discussion

PLO was first described in 1955 by Nordin and Roper [15]. Its incidence is estimated to be 4–8 cases per 1,000,000 women [8,16], but the precise incidence is probably higher due to undiagnosed cases. Back pain caused by a vertebral fracture is one of the most common symptoms of PLO, and more than two-thirds of the vertebral fractures in cases of PLO occur in a first pregnancy [2,17].

Pregnancy and lactation affect bone metabolism. Approximately 200–250 mg of calcium is transferred daily to the fetus and during breastfeeding [1,18]. Most of the calcium used for milk production comes from bone, as women experience a transient 3–9% decrease in bone density during lactation [19]. As in the present patient, lactating women have lower levels of 1.25(OH)_2_D, and higher levels of PTHrP and prolactin compared to those of nonlactating women [20]. Low body weight, vitamin D deficiency, prior low BMD, smoking, excessive alcohol consumption, dysfunction of osteoblasts, and some gene mutations may be predisposing factors for PLO [5,21,22]. Our patient’s mother had been diagnosed with postmenopausal osteoporosis in her 40s, and although this may have been linked to the patient’s premenopausal osteoporosis, the etiology remains unclear.

The goals of treatment for PLO are the prevention of subsequent vertebral and nonvertebral fractures, the relief of pain, and an increase in the patient’s BMD. A mainstay of the treatment for PLO is delactation with supplementation of calcium with or without vitamin D. Calcium and vitamin D supplementation with weaning increased the lumbar BMD value by 3.5–6.2% over a 12-month period [11,23]. Several drugs that are used to treat osteoporosis have also been used for PLO, including bisphosphonate, strontium ranelate (SrRan), denosumab, and teriparatide. To the best of our knowledge, there is no report in the English literature of a patient with PLO treated with romosozumab.

Bisphosphonates increase BMD by inhibiting bone resorption. Several studies observed that patients with PLO treated with alendronate or zoledronate along with calcium and vitamin D supplementation had a 10.2–17.0% annual mean gain in their lumbar spine BMD and a 2.6–6.5% annual mean gain in the femoral neck BMD [2,4,5]. The half-life of a bisphosphonate is as long as 10 years, and bisphosphonates pass through the placenta and may have teratogenic effects on a fetus [24,25,26]. The use of bisphosphonates may, therefore, be risky for the subsequent gestations of PLO patients.

SrRan was used for patients with PLO in limited cases [6,7]. It has a dual effect on bone, i.e., the stimulation of new bone formation and the inhibition of bone resorption [27]. Treatment with SrRan resulted in significant reductions in the risks of new vertebral and nonvertebral fractures, and increased both the lumbar spine and femoral neck BMD in postmenopausal patients [28]. SrRan provides a rapid increase in BMD. Patients with PLO treated with weaning, and a supplementation of SrRan, calcium, and cholecalciferol achieved a 31–33% increase in their lumbar BMD and an almost 20% increase in their total hip BMD at 12–21 months [6,7]. The half-life of SrRan is ~60 h. However, the long-term safety and potential adverse events regarding prenatal impairment that are associated with the use of SrRan are unclear [6].

Denosumab is a human monoclonal antibody that binds to the receptor activator of nuclear factor κB-ligand (RANKL), and inhibits the activation of osteoclasts and their precursors, leading to a suppression of bone turnover and an increase in BMD. The mean half-life of 60 mg denosumab is 25.4 days, and its concentration declines over a period of 4–5 months. Stumpf et al. (2021) reported the case of a patient with PLO treated with denosumab along with calcium and vitamin D supplementation, which led to increases in BMD at the lumbar spine, femoral neck, and total hip by 21.2%, 5.6%, and 8.0% at 12 months and by 32.0%, 13.0%, and 11.5% at 18 months, respectively [8]. Another patient with PLO was treated with weekly teriparatide (56.5 μg/week) for 6 weeks, followed by denosumab (60 mg) for 6 months, and at 12 months, she exhibited increases in BMD from baseline at the lumbar spine (L2–L4) and femoral neck by 16.5% and 3.9%, respectively [9]. These outcomes suggested that denosumab can be considered one of the effective treatments for PLO.

Teriparatide is an osteoanabolic agent and does not accumulate in the bone. Many reports of the BMD increase effect of teriparatide for the treatment of PLO were publicized [8,9,10,11,12,23,29,30,31]. Teriparatide effectively improved the lumbar BMD values of PLO patients by almost 20% at 12 months [11,29] and by 19–36% at 18 months [10,30,31]. The half-life of teriparatide is 1 h, much shorter than the half-lives of bisphosphonates and denosumab. This short half-life of teriparatide suggests that it can be safely used to treat PLO, especially before pregnancy or after weaning. Therefore, we used teriparatide as a first-line treatment. However, the eventual discontinuation of teriparatide treatment poses a problem. In reports from Japan, the 12-month continuation rates of daily, once-weekly, and twice-weekly teriparatide treatments were only 43.1%, 23.5%, and 47.5%, respectively [32,33]. As in our present patient, nausea was cited as one of the major reasons for the difficulty with teriparatide continuation [32,33]. If teriparatide is unavailable because of side effects such as nausea and dizziness, an alternative treatment for PLO is necessary.

Romosozumab is a monoclonal antibody to sclerostin, which is a glycoprotein that inhibits the classic Wnt/β-catenin signaling to suppress bone formation. The half-life of romosozumab is 12.8 days. Romosozumab has the dual effect of increasing bone formation and suppressing bone resorption, leading to significant increases in BMD [34]. In Japan, Inose et al. (2022) recently reported that 106 osteoporosis patients at a high risk of fracture and treated with romosozumab for 12 months achieved BMD increases of 14.6% at the lumbar spine, 5.1% at the femoral neck, and 3.1% at the total femur [35]. Romosozumab was less effective for patients with prior treatment consisting of osteoporosis medication such as bisphosphonate, denosumab, and teriparatide [36,37]. The BMD increase effect was significantly low in the group administered with romosozumab after denosumab administration [36]. In their study of postmenopausal osteoporosis patients, Tominaga et al. (2022) observed that previous treatment with teriparatide did not clearly affect the effectiveness of 12-month romosozumab treatment [38]. We, thus, speculate that, in the present patient, the impact of the prior treatment with teriparatide was not significant.

Saag et al. (2017) reported that the incidence of cardiovascular and cerebrovascular adverse events with romosozumab was increased compared to an alendronate preparation. However, a clinical trial indicated that the rates of severe adverse events appeared to be balanced between patients treated with romosozumab and those who received a placebo [13]. The incidence of adverse events associated with romosozumab treatment remains to be clarified.

In our patient, (i) new vertebral fractures had occurred, (ii) the BMD values had not improved, and (iii) the patient reported feeling severe nausea after every teriparatide injection. Therefore, we judged that the teriparatide treatment did not show beneficial fracture prevention effects on her; thus, the teriparatide treatment was discontinued. However, the patient needed to increase her lumbar spinal BMD quickly to prevent other subsequent vertebral fractures. Romosozumab treatment improved lumbar spinal BMD more quickly than those of teriparatide and denosumab in postmenopausal osteoporotic women [34,39]. Therefore, although there was no previous report of romosozumab treatment for the patient with PLO, we offered the patient to use romosozumab and she agreed with our proposal.

The sequential treatment after anabolic agents is an important concern. In the DATA-Switch study regarding postmenopausal osteoporotic women, switching therapy from teriparatide to denosumab showed a greater increase in BMD at the lumbar spine than that from denosumab to teriparatide [40]. On the other hand, in previous case reports of patients with PLO treated with teriparatide, lumbar spinal BMDs were maintained for more than 14 months after stopping treatment [30,31]. Therefore, after the teriparatide treatment, sequential treatment such as denosumab or bisphosphonate might not be necessary for the patients with PLO.

Our patient’s BMD values increased by 23.6% at the lumbar spine, 4.3% at the femoral neck, and 11.9% at the total hip compared to the baseline without any adverse event; therefore, we suspect that romosozumab may be one of the safe and effective options for reproductive-age patients with PLO. If the present patient develops PLO again after delivering a second child, we plan to treat her with romosozumab again.

In conclusion, 12-month romosozumab treatment after 4-month teriparatide injections increased our PLO patient’s lumbar spine, femoral neck, and total hip BMD values, with no subsequent vertebral fracture. Romosozumab has the potential to be a therapeutic option to improve BMD and reduce the subsequent fracture risk of patients with pregnancy and lactation-associated osteoporosis.

## Figures and Tables

**Figure 1 medicina-59-00019-f001:**
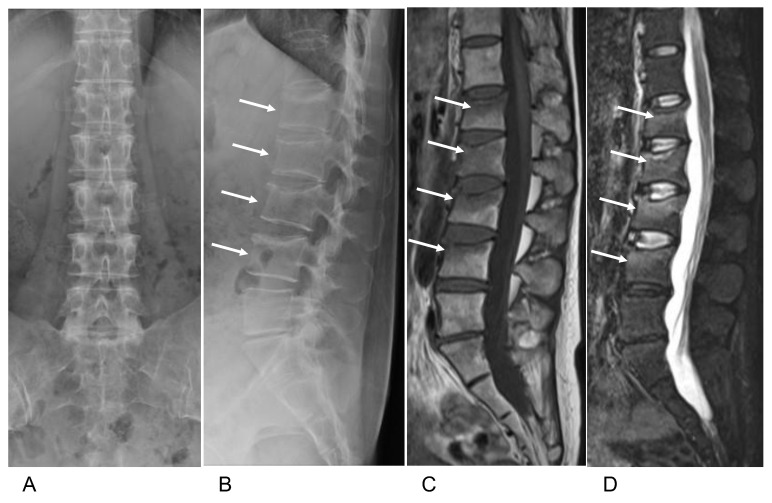
(**A**) Spinal and (**B**) lateral views of the patient’s lumbar spine show loss of vertebral height at L1–L4 (arrow). Sagittal lumbar MRI shows a low-intensity lesion at L1–L4 on T1-weighed imaging (**C**) and a high-intensity lesion on STIR imaging (**D**) (arrow).

**Figure 2 medicina-59-00019-f002:**
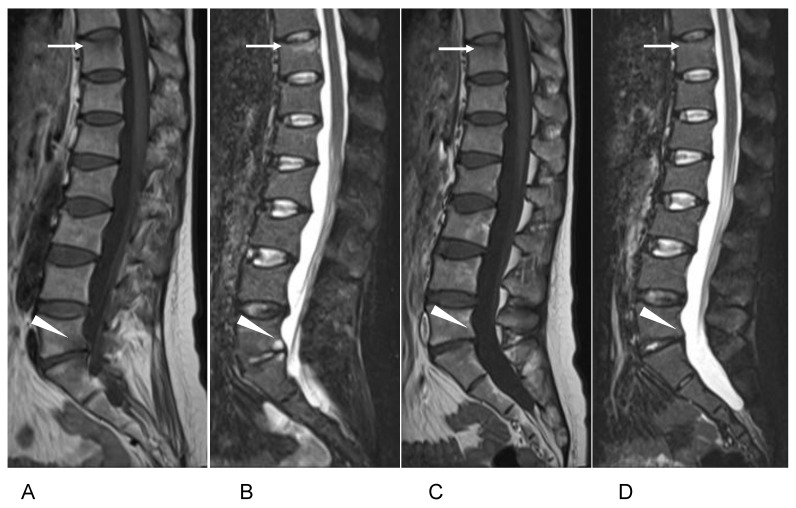
MRI of the patient’s thoracolumbar spine four months after the teriparatide treatment. Sagittal MRI shows low-intensity lesions at Th11 (arrow) and L5 (arrowhead) on (**A**) T1-weighed imaging and (**B**) high-intensity lesions on STIR imaging. Two months after the romosozumab treatment, sagittal MRI showed no abnormal signal change at Th11 (arrow) and L5 (arrowhead) on (**C**) T1-weighed or (**D**) STIR imaging.

**Table 1 medicina-59-00019-t001:** Laboratory test of the patient on first visit, before and after romosozumab treatment.

Value	Normal Range	Before Treatment	Before Romosozumab	After 2 mo	After 6 mo	After 12 mo
TRACP-5b, mU/dL	120–420	476	348	158	96	67
Total PⅠNP, ng/mL	16.8–98.2	76.1	113	71.4	30.7	26.0

TRACP-5b: tartrate-resistant acid phosphatase-5b; PⅠNP: procollagen type Ⅰ N-terminal propeptide.

**Table 2 medicina-59-00019-t002:** Bone mineral density values before and after romosozumab treatment.

	Before Treatment	Before Romo- Sozumab *	2 mo	4 mo	6 mo	8 mo	12 mo	BMD Change at 12 mo vs. Baseline
**Lumbar spine (L1–4):**								
BMD, g/cm^2^	0.852	0.842	0.879	0.920	0.978	1.019	1.053	23.6%
Z score	−2.1	−2.2	−1.9	−1.6	−1.1	−0.8	−0.5	
**Femoral neck:**								
BMD, g/cm^2^	0.710	0.723	0.731	0.711	0.741	0.774	0.754	6.2%
Z score	−1.5	−1.3	−1.3	−1.5	−1.2	−1.1	−1.2	
**Total hip:**								
BMD, g/cm^2^	0.726	0.721	0.770	0.788	0.795	0.797	0.807	11.2%
Z score	−1.7	−1.7	−1.3	−1.2	−1.1	−1.1	−1.0	

* After 4-month bi-weekly teriparatide treatment.

## Data Availability

No new data were created or analyzed in this study. Data sharing is not applicable to this article.

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
