# Peer review of "Pregnancy and Lactation-Associated Osteoporosis Successfully Treated with Romosozumab: A Case Report"

_medicina, 2022, doi:10.3390/medicina59010019_

Round 1

Reviewer 1 Report

The manuscript by Yoichi Kaneuchi et al claimed and concluded that romosozumab has potential as a therapeutic option to improve the BMD and reduce the subsequent fracture risk of patients with PLO. However, the manuscript could be improved if the following comments and questions are addressed.

1.       The authors mentioned that “her PINP level had increased to 113 ng/mL, and TRACP-5b had decreased to 348 mU/dL (Table 1)”. However, I didn’t find the table in the manuscript.

2.       The authors could provide a figure showing the timeline describing change of the patient MRI if possible.

3.       The manuscript is well understandable, however it is not fluent and the language should be improved.

Author Response

Thank you very much for reviewing our manuscript and giving us beneficial advices.

Reviewer 2 Report

1. Insufficient evidence for using anabolic agents such as teriparatide and romosozumab as 1st line treatment of choice for PLO.

2. The effect of previously used teriparatide was not considered

    Osteoporosis drugs have different mechanisms of action and characteristics, and side effects and additional effects of drugs also differ depending on the drug, so when osteoporosis drugs are sequentially applied differently, the effects may appear differently.

In a study that compared and analyzed a group of patients who switched from teriparatide to bisphosphonate or denosumab in cases reporting the results of sequential treatment in osteoporosis treatment, the degree of BMD increase was relatively significant in the group administered with denosumab rather than bisphosphonate. It was higher. In other words, it was reported that denosumab could be a good choice as a bone resorption inhibitor used after teriparatide administration.

  In addition, in a study that analyzed the treatment effects of drugs used before romosozumab treatment, the BMD increase effect was relatively low in the group administered with romosozumab after Denosumab administration.

As such, caution should be exercised in patients who have been administered Denosumab prior to anabolic agent administration, such as careful follow-up of bone density and bone markers.

  Therefore, in this case, it is more important to compare the treatment effect of romosozumab depending on which drug was previously administered as a variable, and to compare with other drugs that may increase the treatment effect other than romosozumab after teriparatide.

3. Vertebra with heterogenous osteolytic bone lesion seen on MRI also seems to require testing to differentiate it from multiple myeloma.

Author Response

(The authors gave the same response as above.)

Round 2

Reviewer 2 Report

1. The patient's recent follow-up MRI view of the spine was helpful in confirming the effect of romosozumab for the ambiguously treated part of the spine .

2. The description of the sequential treatment after anabolic agents and additional explanations and references are relevant in this case